# Prior Cancer Is Associated with Lower Atherosclerotic Cardiovascular Disease Risk at First Acute Myocardial Infarction

**DOI:** 10.3390/biomedicines10112681

**Published:** 2022-10-23

**Authors:** Chieh Yang Koo, Huili Zheng, Li Ling Tan, Ling-Li Foo, Derek J. Hausenloy, Wee-Joo Chng, Soo Chin Lee, Arthur Mark Richards, Lieng-Hsi Ling, Shir Lynn Lim, Chi-Hang Lee, Mark Y. Chan

**Affiliations:** 1Department of Cardiology, National University Heart Centre Singapore, Singapore 119074, Singapore; 2National Registry of Diseases Office, Health Promotion Board, Singapore 168937, Singapore; 3Yong Loo Lin School of Medicine, National University of Singapore, Singapore 119077, Singapore; 4Cardiovascular & Metabolic Disorders Program, Duke-National University of Singapore Medical School, Singapore 169857, Singapore; 5National Heart Research Institute Singapore, National Heart Centre, Singapore 169609, Singapore; 6The Hatter Cardiovascular Institute, University College London, London WC1E6BT, UK; 7Cardiovascular Research Centre, College of Medical and Health Sciences, Asia University, Taichung 41354, Taiwan; 8Department of Haematology-Oncology, National University Cancer Institute of Singapore, Singapore 119074, Singapore; 9Christchurch Heart Institute, University of Otago, Dunedin 9016, New Zealand

**Keywords:** cardio-oncology, coronary artery disease, preventive cardiology, cancer survivorship, risk factors

## Abstract

Background: Patients with cancer are at increased risk of acute myocardial infarction (AMI). It is unclear if the Atherosclerotic Cardiovascular Disease (ASCVD) risk score at incident AMI is reflective of this higher risk in patients with prior cancer than those without. Methods: We linked nationwide AMI and cancer registries from 2008 to 2019. A total of 18,200 eligible patients with ASCVD risk score calculated at incident AMI were identified (1086 prior cancer; 17,114 no cancer). Results: At incident AMI, age-standardized mean ASCVD risk was lower in the prior cancer group (18.6%) than no cancer group (20.9%) (*p* < 0.001). Prior to incident AMI, smoking, hypertension, hyperlipidemia and diabetes mellitus were better controlled in the prior cancer group. However post-AMI, prior cancer was associated with lower guideline-directed medical therapy usage and higher all-cause mortality (adjusted hazard ratio 1.85, 95% confidence interval 1.66–2.07). Conclusions: AMI occurred despite better control of cardiovascular risk factors and lower age-standardized estimated mean 10-year ASCVD risk among patients with prior cancer than no cancer. Prior cancer was associated with lower guideline-directed medical therapy post-AMI and higher mortality.

## 1. Introduction

Cancer survivorship has improved greatly due to advancements in cancer detection and therapy [1]. Although cancer-related outcomes have improved, cancer survivors remain at a significantly higher risk of cardiovascular disease, including acute myocardial infarction (AMI) [2,3]. In certain cancers, such as prostate cancer, the risk of cardiovascular mortality has even been reported to be higher than the risk cancer mortality [4]. This greater risk of cardiovascular disease and AMI has been attributed to several factors, including shared risk factors between cancer and atherosclerosis, possibly operating through common biological pathways such as chronic inflammation and clonal hematopoiesis of indeterminate potential, as well as iatrogenic factors such as cardiotoxic effects from cancer therapies [5,6,7,8].

Existing strategies for primary prevention of cardiovascular disease include the routine use of scores for risk stratification and guidance of treatment. The well-established Atherosclerotic Cardiovascular Disease (ASCVD) risk calculator endorsed by the American College of Cardiology is one of the most commonly used risk scores for primary prevention of cardiovascular disease [9]. However, this score was originally intended for the general population, and recent guidelines have identified cancer survivors as a specific population with knowledge gaps in the primary prevention of cardiovascular disease [10]. The applicability of the ASCVD score to cancer survivors is therefore uncertain. Recommendations from oncology guidelines advocate screening and treatment of cardiovascular risk factors for primary prevention in cancer survivors but do not specify clear treatment targets [11,12]. We have previously reported lower serum cholesterol concentrations, an important variable in existing risk scores including the ASCVD score, in patients with cancer compared with non-cancer controls at index AMI and stroke [13]. Hence, we hypothesized that patients with prior cancer would have a lower ASCVD score at incident AMI compared with those without cancer.

To better understand the association of ASCVD scores in patients with prior cancer, we combined data across national registries to calculate the ASCVD risk scores at incident AMI, comparing risk scores in those with prior cancer to those without. Our objectives were first to determine if the age-standardized ASCVD score calculated at incident AMI was lower in patients with prior cancer than those without cancer, and second, to compare treatment of cardiovascular risk factors prior to incident AMI between the two groups of patients. We hypothesized that prior cancer was associated with lower ASCVD scores and a lower rate of treatment of cardiovascular risk factors prior to incident AMI.

## 2. Materials and Methods

### 2.1. Study Design and Participants

In this population-based cohort study, we combined data from two national registries managed by the National Registry of Diseases Office of Singapore [14]. The National Registry of Disease Act mandates notification of all AMI and cancer cases within Singapore. The Singapore Myocardial Infarction Registry consists of data from patients diagnosed with AMI from 2007 across all hospitals within the country. Specialized coordinators collected data including patient demographics, clinical presentation, medications, laboratory and procedural parameters [15]. As individual-level blood pressure values were only available within the registry from 2017, data on blood pressure was extracted from hospital-level databases from two centers and merged with the national registry for the years 2007 to 2016. The Cancer Registry consists of data from patients diagnosed with cancer from 1968 across all hospitals within the country. Data captured included patient demographics, cancer characteristics and treatment details [16]. The local institutional review board (Domain Specific Review Board-C, National Healthcare Group, 2021/00141) approved this study with waiver of consent.

Data between the Singapore Myocardial Infarction Registry and the Cancer Registry were linked to identify patients with prior cancer presenting with AMI over the 12 years from 1 January 2008 to 31 December 2019. The Registry of Birth and Death was then linked with this combined dataset to ascertain mortality. As the ASCVD score is used for primary prevention and is not applicable for secondary prevention, we excluded patients with a prior history of AMI, stroke, percutaneous coronary intervention or coronary artery bypass grafting surgery. We also excluded patients who developed cancer after incident AMI, as well as patients with incomplete lipid or blood pressure data, thus preventing complete calculation of ASCVD scores.

Each individual had their predicted risk of cardiovascular disease calculated via the ASCVD risk score via data obtained just prior to or at incident AMI. The ASCVD estimates a 10-year composite risk of coronary death, non-fatal AMI and fatal or non-fatal stroke. It incorporates nine variables: age, sex, race, diabetes, smoking, total cholesterol, high density lipoprotein cholesterol (HDL-C), systolic blood pressure and treatment for hypertension. Individuals are classified as low-risk (ASCVD risk score < 5% 10-year risk), borderline-risk (ASCVD risk score 5% to <7.5% 10-year risk), intermediate-risk (ASCVD risk score ≥ 7.5% to <20% 10-year risk) and high-risk (ASCVD risk score ≥ 20% 10-year risk) [17].

The primary outcome of interest was the ASCVD risk score at incident AMI. Secondary outcomes of interest included rates of treatment of cardiovascular risk factors prior to incident AMI and post-AMI mortality. AMI patients were classified into two groups: patients with prior cancer (prior cancer) and patients without prior cancer (no cancer). We excluded all missing data from the analyses via case deletion without imputation. STATA SE version 13 was used to perform all statistical analyses.

### 2.2. Statistical Analysis

Demographics, comorbidities and clinical characteristics at incident AMI were compared between those with prior cancer and those without using a chi-square test for categorical variables and Wilcoxon rank-sum test for numeric variables. As the patients with prior cancer were older than those without cancer, we standardized their age structure to those without cancer (i.e., using the age structure of patients without cancer as reference). We compared the risk distribution between the two groups by the Chi-square test. Among those with prior cancer, we further categorized risk distribution by cancer characteristics. Kaplan–Meier curves were plotted to compare the cumulative incidence of all-cause death during the study period between the two groups of patients. The hazard ratio of all-cause death among those with prior cancer compared with those without cancer was assessed by Cox regression adjusted for age, sex, smoking status, hypertension, hyperlipidemia and diabetes mellitus.

## 3. Results

### 3.1. Baseline Demographics

We identified 90,297 patients with AMI between 1 January 2008 and 31 December 2019. Following the exclusion of patients with a history of prior AMI, stroke, percutaneous coronary intervention or coronary artery bypass grafting surgery, 71,377 patients with incident AMI without prior cardiovascular disease remained. We then further excluded patients who developed cancer after incident AMI and patients with incomplete data for calculation of ASCVD risk score. A total of 18,200 patients were available for final analysis, of which 1086 patients had prior cancer (prior cancer group) and 17,114 patients did not (no cancer group) (Figure 1). The median duration from diagnosis of cancer to AMI was 2707 days. Table 1 presents the baseline characteristics of both groups. The prior cancer group was older, with a lower proportion of men, more patients of Chinese ethnicity and lower body mass index.

At incident AMI, the prior cancer group had fewer current smokers (Table 1). The prevalence of hypertension was higher among the prior cancer group, with more patients on treatment for hypertension, and significantly lower systolic and diastolic blood pressures were observed at incident AMI. Although the prevalence of hyperlipidemia did not differ significantly between both groups, more patients within the prior cancer group were on treatment for hyperlipidemia. At incident AMI, the prior cancer group had significantly higher median concentrations of high-density lipoprotein cholesterol, and lower median concentrations of total cholesterol and low-density lipoprotein cholesterol. Similarly, although the prevalence of diabetes mellitus did not differ significantly between both groups, more patients within the prior cancer group were on treatment for diabetes mellitus with a lower median glycated hemoglobin at incident AMI.

There were fewer ST-segment elevation myocardial infarctions in the prior cancer group, and these patients had lower rates of revascularization (Table 1). Although there was a higher incidence of heart failure in the prior cancer group, there was no significant difference in the incidence of cardiogenic shock or median left ventricular ejection fraction. The prior cancer group had lower rates of guideline directed medical therapy prescribed at discharge after AMI.

### 3.2. Variation in Predicted Cardiovascular Risk by Cancer Status

The median ASCVD score for the entire study cohort predicted a 14.8% 10-year risk of cardiovascular events. In unadjusted analyses, the prior cancer group had a significantly higher median predicted ASCVD risk and a higher proportion of high ASCVD risk than the no cancer group, reflecting the greater age of the cancer group. Table 2 shows the variation in ASCVD risk score by cancer status. In patients aged 40 to 70 years, prior cancer was instead associated with a lower median ASCVD risk compared with no cancer. When standardized by age, the mean-predicted ASCVD risk was significantly lower in prior cancer (18.6%) than no cancer (20.9%). Figure 2 shows the distribution of predicted ASCVD risk by cancer status and the lower mean-predicted ASCVD risk after age standardization.

### 3.3. Post-AMI Mortality by Cancer Status

Figure 3 demonstrates an increased risk of post-AMI mortality among the prior cancer group (unadjusted hazard ratio 3.43, 95% confidence interval 3.08–3.82) compared with the no cancer group. After adjusting for age, sex, smoking status, hypertension, hyperlipidemia and diabetes mellitus, the risk of post-AMI mortality in the prior cancer group remained significantly greater (adjusted hazard ratio 1.85, 95% confidence interval 1.66–2.07).

## 4. Discussion

This study reports on the differences in estimated cardiovascular risk via the ASCVD risk score among a national cohort of patients with and without prior cancer presenting with their first AMI. Amongst more than 18,000 patients with AMI over 12 years, the age-standardized mean 10-year ASCVD risk was 18.6% in the prior cancer group, significantly lower than the 20.9% observed in the no cancer group. Prior cancer patients were more likely to be receiving treatment for their cardiovascular risk factors prior to incident AMI but reported lower use of guideline-directed medical therapy post-AMI than the no cancer group. Prior cancer was associated with lower post-AMI survival than no cancer.

The main finding of this study is that AMI occurred despite a lower age-standardized estimated mean 10-year ASCVD risk in prior cancer than no cancer. This suggests that while traditional cardiovascular risk factors are better controlled in patients with prior cancer, there may be other non-traditional risk factors contributing to AMI risk unaccounted for by the ASCVD risk score. Hence, there remain clear limitations when applying the ASCVD risk prediction score to cancer patients which may underestimate the ASCVD risk. This is likely due to several potential non-traditional risk factors which may increase the risk of adverse cardiovascular events in patients with prior cancer—first, cancer therapies are commonly associated with cardiotoxicity in nearly a third of all patients [5]. These include well-established agents such as 5-fluorouracil and radiotherapy, and even newer agents, such as immune checkpoint inhibitors. Of note, immune checkpoint inhibitors have been associated with the progression of atherosclerotic plaque volume and increased risk of AMI [18]. This unaccounted impact of potentially cardiotoxic cancer therapies within the ASCVD score could account for our findings of prior cancer patients developing AMI despite lower ASCVD scores at incident AMI. Several risk scores incorporating cancer treatment variables have been proposed for use within specific cancer subtypes such as breast cancer, but these still require further validation before routine clinical use [19,20]. Second, common pathobiological mechanisms underpinning both cancer and atherosclerosis have been identified. For example, the role of inflammation is increasingly recognized as a common driver of both cancer and atherosclerosis onset and progression [21,22]. Treatment with a human monoclonal antibody against the inflammatory cytokine interleukin-1β reduced the risk of AMI and incident lung cancer [8,23]. Additionally, clonal hematopoiesis of indeterminate potential has been associated with increased risks of both hematologic cancer and AMI [7]. Accounting for these less traditional cardiovascular risk factors may enhance risk stratification in cardiovascular primary prevention among cancer patients.

We also observed that patients with prior cancer had their traditional cardiovascular risk factors better controlled prior to incident AMI, yet were discharged with lower rates of guideline-directed medical therapy post-AMI compared with patients without cancer. The improved control of cardiovascular risk factors prior to AMI could be due to patients with prior cancer having greater contact with healthcare professionals compared with patients without cancer. Conversely, the low rates of guideline-directed medical therapy in patients with prior cancer compared with patients without cancer is likely due to treatment bias. Studies have demonstrated lower prescription of ideal cardiovascular health behaviors and therapies to patients with cancer [24,25]. Greater awareness is required to optimize post-AMI therapy in this high-risk population to improve cardiovascular outcomes. A dedicated cardio-oncology service with clinicians more aware of these treatment biases may help to improve cardiovascular outcomes [26]. This is especially pertinent, as our findings showed that prior cancer was associated with higher post-AMI mortality than no cancer, which is also consistent with multiple other studies [2,27].

Our findings highlight that clear treatment targets for primary prevention of cardiovascular disease in cancer patients are required. Although our findings demonstrated that the traditional cardiovascular risk factors were better controlled prior to incident AMI within the prior cancer group than the no cancer group, the intensity of therapy was lower than expected for the associated ASCVD score. This further highlights the current evidence gap in which current guidelines on primary prevention do not specifically provide treatment targets for patients with cancer [10,11,12]. Our prior findings that patients with prior cancer at incident AMI have lower serum cholesterol concentrations than patients without cancer may suggest that existing guidelines based on the general population are inadequate when extrapolated to the prior cancer population [13]. The American College of Cardiology have proposed several conditions to be considered as atherosclerotic cardiovascular disease risk enhancers, including inflammatory disease such as rheumatoid arthritis or human immunodeficiency virus infection [28]. Established risk scores have also been shown to underestimate cardiovascular risk within these specific patient populations [2,3,4,5,6,7,8,9,10,11,12,13,14,15,16,17,18,19,20,21,22,23,24,25,26,27,28,29,30,31]. Hence, we propose that further research is needed to ascertain the value of adding cancer as a variable to the ASCVD score, as well as to identify and include relevant cancer-related variables to improve its applicability within patients with prior cancer.

There are several limitations in our study. First, the ASCVD was developed primarily for use within the American population and may not be fully applicable to the multi-ethnic Singaporean population. Second, the ASCVD is meant for primary prevention in patients older than 40 years old, but we included patients younger than 40 years due to the importance of risk stratification within childhood cancer survivors who are equally of high cardiovascular risk [32]. Finally, the Cancer Registry alone had limited data, which precluded the calculation of the ASCVD risk score for all patients with cancer. Hence, we were unable to form a control group of cancer patients without AMI within the Cancer Registry. We were also unable to compare expected versus observed rates of AMI within the entire cancer population.

## 5. Conclusions

In conclusion, AMI occurred despite better control of traditional cardiovascular risk factors and a lower age-standardized estimated mean 10-year ASCVD risk among patients with prior cancer than patients with no cancer. Patients with prior cancer have better control of traditional cardiovascular risk factors before AMI but less guideline-directed medical therapy after AMI and higher post-AMI mortality. Additional research to support appropriate recommendations for best clinical practice are needed to improve the prevention and management of cardiovascular disease among cancer survivors.

## Figures and Tables

**Figure 1 biomedicines-10-02681-f001:**
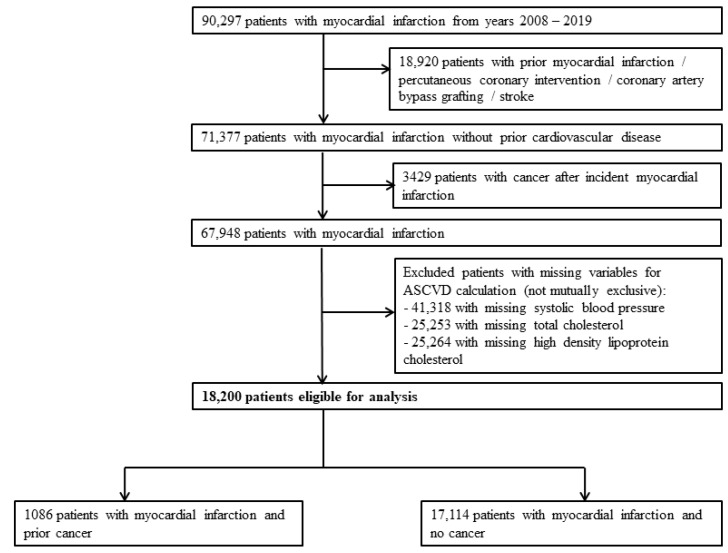
Study design and patient selection flow chart.

**Figure 2 biomedicines-10-02681-f002:**
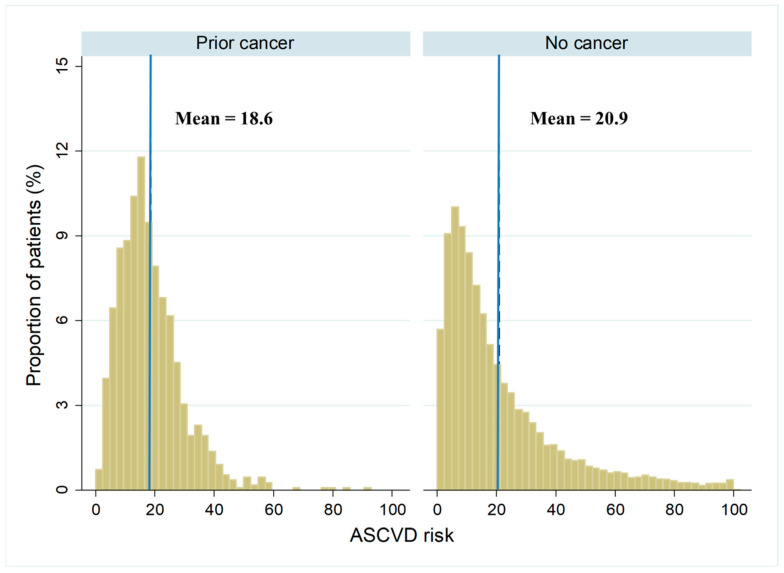
Distribution of age-standardized ASCVD risk by cancer status. The distribution of ASCVD risk classified according to cancer status after adjusting the age structure of the prior cancer group to the no cancer group with mean ASCVD risk indicated by vertical line. ASCVD: atherosclerotic cardiovascular disease.

**Figure 3 biomedicines-10-02681-f003:**
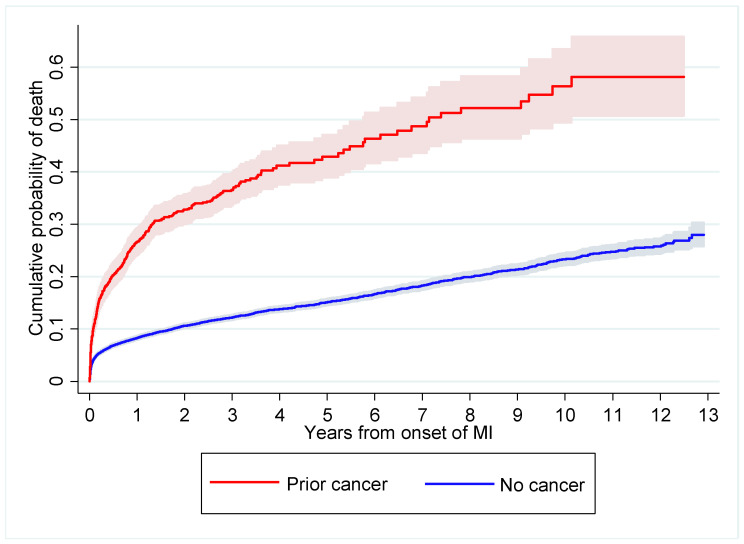
Post-AMI mortality according to cancer status. The red line represents the prior cancer group while the blue line represents the no cancer group. Confidence interval indicated by shaded area surrounding line. AMI: acute myocardial infarction.

**Table 1 biomedicines-10-02681-t001:** Baseline patient demographics and clinical presentation.

	Prior Cancer (*n* = 1086)	No Cancer (*n* = 17,114)	*p* Value
Age, median (IQR), years	73 (64–81)	61 (53–70)	<0.001
Age group, n (%)			
<40	7 (0.6)	532 (3.1)	
40–49	34 (3.1)	2454 (14.3)	
50–59	124 (11.4)	5106 (29.8)	
60–69	295 (27.2)	4784 (28.0)	
70–79	312 (27.2)	2631 (15.4)	
80–89	272 (25.1)	1351 (7.9)	
>=90	42 (3.9)	256 (1.5)	
Male sex, n (%)	641 (59.0)	13,293 (77.7)	<0.001
Ethnicity, n (%)			<0.001
Chinese	873 (80.4)	10,809 (63.2)	
Malay	111 (10.2)	3685 (21.5)	
Indian	87 (8.0)	2349 (13.7)	
Others	15 (1.4)	271 (1.6)	
Body mass index, median (IQR), kg/m^2^	23.2 (20.5–26.0)	24.7 (22.4–27.6)	<0.001
Cardiovascular risk factors, *n* (%)			
Current smoker	161 (14.8)	6677 (39.0)	<0.001
Hypertension	771 (80.0)	9852 (57.6)	<0.001
Received treatment for hypertension	608 (56.0)	6885 (40.2)	<0.001
Systolic blood pressure, median (IQR), mmHg	133 (114–153)	135 (117–155)	0.008
Diastolic blood pressure, median (IQR), mmHg	73 (62–85)	79 (68–92)	<0.001
Hyperlipidaemia	673 (62.0)	10,713 (62.6)	0.679
Received treatment for hyperlipidaemia	427 (39.3)	5131 (30.0)	<0.001
Total cholesterol, median (IQR), mmol/L	4.20 (3.38–5.16)	4.91 (4.10–5.80)	<0.001
HDL cholesterol, median (IQR), mmol/L	1.10 (0.90–1.38)	1.08 (0.90–1.29)	0.006
LDL cholesterol, median (IQR), mmol/L	2.50 (1.73–3.33)	3.15 (2.40–3.97)	<0.001
Diabetes mellitus	427 (39.3)	6497 (38.0)	0.372
Received treatment for diabetes mellitus	304 (28.0)	3914 (22.9)	<0.001
HbA1c, median (IQR), %	5.9 (5.5–6.9)	6.0 (5.6–7.4)	<0.001
MI characteristics, *n* (%)			
STEMI	337 (33.5)	7770 (46.8)	<0.001
Underwent revascularization	544 (50.1)	12,591 (73.6)	<0.001
LVEF, median (IQR), %	50 (35–60)	50 (38–59)	0.587
Complications, *n* (%)			
Cardiac arrest presentation	12 (1.1)	330 (1.9)	0.053
Heart failure	81 (7.5)	982 (5.7)	0.019
Cardiogenic shock	23 (2.1)	340 (2.0)	0.761
* Stent thrombosis	0 (0.0)	32 (0.2)	0.139
Medications at discharge, *n* (%)			
Aspirin	833 (85.0)	15,368 (93.3)	<0.001
Beta-blocker	777 (79.3)	13,775 (83.6)	<0.001
ACE-I/ARB	538 (54.9)	10,975 (66.6)	<0.001
Lipid lowering therapy	910 (92.9)	15,975 (97.0)	<0.001

Abbreviations: ACE-I: angiotensin-converting enzyme inhibitor; ARB: angiotensin II receptor blocker; HDL: high-density lipoprotein cholesterol; IQR: interquartile range; LDL: low-density lipoprotein cholesterol; LVEF: left ventricular ejection fraction; MI: myocardial infarction; STEMI: ST-segment elevation myocardial infarction. Unknown values were excluded. * Data on thrombosis are only available from 2012 onwards.

**Table 2 biomedicines-10-02681-t002:** ASCVD risk by cancer status.

	Prior Cancer (*n* = 1086)	No Cancer (*n* = 17,114)	*p* Value
Predicted ASCVD risk, median (IQR), %			
Age <40 years	1.6 (0.3–3.3)	2.5 (1.1–5.0)	0.285
Age 40–49 years	3.8 (2.2–6.9)	5.4 (2.8–9.0)	0.048
Age 50–59 years	5.8 (3.3–10.0)	9.7 (6.0–15.1)	<0.001
Age 60–69 years	15.0 (8.9–22.3)	16.9 (11.1–24.9)	<0.001
Age 70–79 years	29.0 (21.1–41.4)	30.9 (21.8–43.3)	0.107
Age 80–89 years	57.0 (41.4–70.8)	56.9 (42.2–72.2)	0.610
Age >=90 years	84.4 (69.8–94.0)	87.1 (72.2–96.9)	0.330
* Predicted age-standardized ASCVD risk, mean (SD), %	18.6 (11.0)	20.9 (19.7)	<0.001

Abbreviations: ASCVD: atherosclerotic cardiovascular disease; IQR: interquartile range; SD: standard deviation. * Standardized to the age distribution of the no cancer group. Mean used for standardization purposes. Median was used prior as the distribution of ASCVD score is skewed for both the prior cancer and no cancer groups.

## Data Availability

The data presented in this study are available on request from the corresponding author.

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
