# Peer review of "Prior Cancer Is Associated with Lower Atherosclerotic Cardiovascular Disease Risk at First Acute Myocardial Infarction"

_biomedicines, 2022, doi:10.3390/biomedicines10112681_

Round 1

Reviewer 1 Report

Excellent read. I would suggest including in your discussion the potential benefits of having a cardio-oncology service to mitigate the "treatment paradox"- clinicians who are more aware of the  cardiotoxiticy/late effects from cancer treatment and therefore better able to manage this cohort of patients. 

Also, why were prior cancer patients less likely to be revascularised? 

Author Response

Response to Reviewer 1 Comments

Point 1: Excellent read. I would suggest including in your discussion the potential benefits of having a cardio-oncology service to mitigate the "treatment paradox"- clinicians who are more aware of the  cardiotoxiticy/late effects from cancer treatment and therefore better able to manage this cohort of patients. 

Response 1: Thank you for your very kind and favorable review. We have added into our discussion about the potential role of cardio-oncology in mitigating this “treatment paradox” as per your kind suggestion.

“A dedicated cardio-oncology service with clinicians more aware of these treatment biases may help to improve cardiovascular outcomes.” [1] (Lines 271-273)

  1. Pareek N, Cevallos J, Moliner P, Shah M, Tan LL, Chambers V, Baksi AJ, Khattar RS, Sharma R, Rosen SD, Lyon AR. Activity and outcomes of a cardio-oncology service in the United Kingdom-a five-year experience. Eur J Heart Fail. 2018;20(12):1721-1731.

Point 2: Also, why were prior cancer patients less likely to be revascularised? 

Response 2: This is an important question. Many studies have shown similar findings where cancer patients were less likely to be revascularised. This could be due to treatment biases where such patients were more likely to be older or have more comorbidities [1]. Prior cancer patients receiving active treatment may also be at increased risk from bleeding complications or have an unclear prognosis and hence treating physicians may be less inclined to pursue revascularisation aggressively.

  1. Leedy D, Tiwana JK, Mamas M, et al. Coronary revascularisation outcomes in patients with cancer. Heart 2022;108:507-516.

Reviewer 2 Report

In their paper “Prior Cancer is Associated with Lower Atherosclerotic Cardiovascular Disease Risk at First Acute Myocardial Infarction” the authors deal with an important clincal question, i.e. better understanding the cardiovascular risk of cancer patients. The main conclusion supported by the data presented is that among younger patients with prior cancer, the ASCVD score may underestimate the risk of AMI

While the study is of interest and technically sound, there are serious conceptual concerns related to the paper. These concerns are related to conclusions drawn by the authors not supported by their data.

Specifically: Figure 2 and Figure 3 are interesting

BUT:

Table 3 does not make any sense. What are the authors trying to show by this analysis?

Major issues:

Research question

Phrasing of research question line 19 and 20 unclear. In contrast, in the prior study of the authors (Koo et al. Sci Rep 2021), the question is phrased clearly and is clinically sound.

Line 66:

Hypothesis: Why is a higher ASCVD score predicted when previous data from the authors themselves (reference 13) suggests the opposite?

The value of ASCVD score would be possible to assess when patients suffering from cardiovascular events within 10 years were compared to those without any events. As this is not the design of the study, thre research question needs t be rephrased.

Study suffers from serious pre-selection bias

ASCVD score is a tool of primary prevention. It should have been calculated for patients with MACE as a readout in a longitudinal fashion. The significance of calculating ASCVD score at the onset of AMI is unclear and not applicable to clinical practice.

Following from this conceptual difficulty, there are a number of issues:

Claims in 292 is not viable, in my opinion, as only AMI patients were included into the study. Thus, the comparison of ASCVD score is valid but no claim about the value of ASCVD because of this pre-selection bias.

The authors mention this limitation themselves in line 303. I believe, this limitation is serious!

Line 95:

Exclusion of patients is problematic. This study reduced the valuable database of the Sigapore Registry to a very specific subgroup. i.e. patients with prior cancer and first-onset AMI. The relationship between cancer of cardiovascular disease is not captured reasonably in this way. In this way, about 90% of the available patients are excluded.

Singapore Myocardial Infarction Registry is a valuable data source! Should be made use of s efficiently as possible.

Conclusions drawn

Lines 193 following:

Problematic Passage “The 193 longer the interval from diagnosis of prior cancer, the higher the proportion of patients classified as high risk. Only slightly over a quarter of those aged younger than 40 years at cancer diagnosis who developed incident AMI were identified as high-risk.”

This suggests some kind of relationship, which misinterprets the scope of ASCVD risk score. The relationships described here are simply due to the different age and the onset of CV-risk factors depending on age is a well-known and not at all new phenomenon.

The authors treat ASCVD risk as if it was a biological trait (i.e. line 228). This is not the case as it is simply a score calculated from clinical parameters of patients. Therefore, higher age results in higher ASCVD score on average. The analyses relating to this interrelationship are misleading. Age is a parameter used for calculating ASCVD!

Analyses discussed in line 228 and 229 do not refer to age-control

On the same page, sentence in line 239-240 is problematic, line 260, again!  

On the same page, in Table 1 it becomes apparent that groups are very different from each other. This is problematic for the overall relevance of the study.

Minor issues:

„post-AMI treatment“ line 32

What does that mean?

Figure 1: Transition from events to patients. Needs to be clarified

Paradoxical findings discussed in line 266 should be formulated much more cautiously. Is the analysis powered for the comparison of treatment regimens?

Line 123: What event free survival (what events and in what time frame)? Is quite unclear

Conclusion:

Overall, these are interesting, yet not entirely novel data. Nonetheless, from a clinical point of view there is a point in the analysis. However, the purpose of the investigation needs to be phrased much more concisely in order not to raise claims that cannot be met by the study.

In its current form, the study simply states, that cancer increases the risk of patients to suffer from cardiovascular events, which is not captured by the ASCVD score. This is not a new finding but not an unimportant one, neither. The missed chance of the authors seems to be that they do not address a clinically urgent topic, i.e. to make a suggestions for the modification of the ASCVD risk score to incorporate cancer. (see Discussion in line 237 and line 279

Author Response

Response to Reviewer 2 Comments

Point 1: In their paper “Prior Cancer is Associated with Lower Atherosclerotic Cardiovascular Disease Risk at First Acute Myocardial Infarction” the authors deal with an important clincal question, i.e. better understanding the cardiovascular risk of cancer patients. The main conclusion supported by the data presented is that among younger patients with prior cancer, the ASCVD score may underestimate the risk of AMI

While the study is of interest and technically sound, there are serious conceptual concerns related to the paper. These concerns are related to conclusions drawn by the authors not supported by their data.

Specifically: Figure 2 and Figure 3 are interesting

BUT:

Table 3 does not make any sense. What are the authors trying to show by this analysis?

Response 1: Thank you for stating that the study is of interest and technically sound. However, we do note your conceptual concerns and we aim to address all your concerns. In Table 3, we were initially attempting to demonstrate the difference in ASCVD scores in patients with cancer grouped according to the various cancer characteristics. However, we acknowledge that the study is not powered to detect such differences and hence may appear misleading. As such, we have removed Table 3 from the manuscript and its associated findings.

Point 2: Research question

Phrasing of research question line 19 and 20 unclear. In contrast, in the prior study of the authors (Koo et al. Sci Rep 2021), the question is phrased clearly and is clinically sound.

Line 66:

Hypothesis: Why is a higher ASCVD score predicted when previous data from the authors themselves (reference 13) suggests the opposite?

The value of ASCVD score would be possible to assess when patients suffering from cardiovascular events within 10 years were compared to those without any events. As this is not the design of the study, thre research question needs t be rephrased.

Response 2: Thank you for your comments regarding our research question. As you have rightly pointed out, our previous data suggests that prior cancer patients may have a lower ASCVD score given the findings of lower cholesterol levels in prior cancer patients at incident myocardial infarction or stroke. We agree with this incongruence and thank you for the suggestion. We have added the following to refine our research question and hypothesis.

“It is unclear if the Atherosclerotic Cardiovascular Disease (ASCVD) risk score at incident AMI is reflective of this higher risk in patients with prior cancer than those without.”

“We hypothesized that patients with prior cancer had a lower ASCVD score at incident AMI compared to those without cancer.”

Point 3: Study suffers from serious pre-selection bias

ASCVD score is a tool of primary prevention. It should have been calculated for patients with MACE as a readout in a longitudinal fashion. The significance of calculating ASCVD score at the onset of AMI is unclear and not applicable to clinical practice.

Following from this conceptual difficulty, there are a number of issues:

Claims in 292 is not viable, in my opinion, as only AMI patients were included into the study. Thus, the comparison of ASCVD score is valid but no claim about the value of ASCVD because of this pre-selection bias.

The authors mention this limitation themselves in line 303. I believe, this limitation is serious!

Response 3: We agree that this a serious limitation, and that the significance of calculating the ASCVD score at onsent of AMI is unclear. Unfortunately, our analysis was limited by the existing data available with the registries. Hence, we have decided to phrase our findings more clearly; from evaluating the value of the ASCVD score as a tool for primary prevention, to describing the ASCVD score at incident AMI as a possible surrogate for cardiovascular risk factor status. Hence, we are more cautious and will rephase our claims and findings as subsequently highlighted in your comments. As suggested, we have removed the claim in 292 regarding the value of the ASCVD.

Point 4: Line 95:

Exclusion of patients is problematic. This study reduced the valuable database of the Sigapore Registry to a very specific subgroup. i.e. patients with prior cancer and first-onset AMI. The relationship between cancer of cardiovascular disease is not captured reasonably in this way. In this way, about 90% of the available patients are excluded.

Singapore Myocardial Infarction Registry is a valuable data source! Should be made use of s efficiently as possible.

Response 4: Thank you for highlighting that the Singapore Myocardial Infarction Registry (SMIR) is indeed a valuable data source, and we are in the midst of further analyses for potential subsequent publications. This study actually included all patients in the SMIR regardless if they had cancer or not, but the majority of such exclusions were due to missing data (especially blood pressure) in the earlier years. We also excluded patients with any prior AMI because the ASCVD would not be applicable in such patients. Nevertheless, we do acknowledge that there were many patients excluded due to missing variables.

Point 5: Lines 193 following:

Problematic Passage “The 193 longer the interval from diagnosis of prior cancer, the higher the proportion of patients classified as high risk. Only slightly over a quarter of those aged younger than 40 years at cancer diagnosis who developed incident AMI were identified as high-risk.”

This suggests some kind of relationship, which misinterprets the scope of ASCVD risk score. The relationships described here are simply due to the different age and the onset of CV-risk factors depending on age is a well-known and not at all new phenomenon.

The authors treat ASCVD risk as if it was a biological trait (i.e. line 228). This is not the case as it is simply a score calculated from clinical parameters of patients. Therefore, higher age results in higher ASCVD score on average. The analyses relating to this interrelationship are misleading. Age is a parameter used for calculating ASCVD!

Response 5: We agree, and as per the previous suggestion regarding Table 3, we have removed the entire section on Table 3 including from the results and the discussion. We have also removed the statements in line 228 from the discussion completely. We are now more cautious with the conclusions drawn from our findings.

Point 6: Analyses discussed in line 228 and 229 do not refer to age-control

Response 6: As above, we agree with your opinion and have since removed these statements (line 228 and 229) from the discussion.

Point 7: On the same page, sentence in line 239-240 is problematic, line 260, again!  

Response 7: Thank you for your suggestions. We have made the following amendments.

“This is likely due to several potential non-traditional risk factors which may increase the risk of adverse cardiovascular events in patients with prior cancer”

“Accounting for these less traditional cardiovascular risk factors may enhance risk stratification in cardiovascular primary prevention among cancer patients”

Point 8: On the same page, in Table 1 it becomes apparent that groups are very different from each other. This is problematic for the overall relevance of the study.

Response 8: This is indeed a limitation of the study design. We acknowledge that the groups are different from one another. However, we feel that this reflects real-world data and we hope to be able to describe how patients with prior cancer are different from patients without cancer. We also attempted to mitigate this limitation by adjusting our post-AMI mortality analysis for important variables including age, sex, smoking, hypertension, hyperlipidaemia and diabetes mellitus. In view of this limitation, we have been cautious when making recommendations based on our findings.  

Point 9: „post-AMI treatment“ line 32

What does that mean?

Response 9: Thank you for highlighting. We have clarified the statement as below.

“Prior cancer was associated with lower guideline directed medical therapy post-AMI and higher mortality.”

Point 10: Figure 1: Transition from events to patients. Needs to be clarified

Response 10: Thank you for highlighting. We have edited Figure 1 to display the data as patients only. The new Figure 1 is as below.

Point 11: Paradoxical findings discussed in line 266 should be formulated much more cautiously. Is the analysis powered for the comparison of treatment regimens?

Response 11: Thank you for this suggestion. Our analysis is not powered for the comparison of treatment regimens, and hence we have remove the term “treatment paradox” and have rephased our discussion more cautiously as suggested.

“We also observed that patients with prior cancer had their traditional cardiovascular risk factors better controlled prior to incident AMI, but yet were discharged with lower rates of guideline directed medical therapy post-AMI compared to patients without cancer. The improved control of cardiovascular risk factors prior to AMI could be due to patients with prior cancer having greater contact with healthcare professionals compared to patients without cancer. Conversely, the low rates of guideline directed medical therapy in patients with prior cancer compared to patients without cancer is likely due to treatment bias. Studies have demonstrated lower prescription of ideal cardiovascular health behaviours and therapies to patients with cancer [25, 26].”

Point 12: Line 123: What event free survival (what events and in what time frame)? Is quite unclear

Response 12: Thank you for pointing this uncertainty out. We have rephrased the sentence to the following for greater clarity:

”Kaplan-Meier curves were plotted to compare the cumulative incidence of all-cause death during the study period between the two groups of patients.”

Point 13: Overall, these are interesting, yet not entirely novel data. Nonetheless, from a clinical point of view there is a point in the analysis. However, the purpose of the investigation needs to be phrased much more concisely in order not to raise claims that cannot be met by the study.

In its current form, the study simply states, that cancer increases the risk of patients to suffer from cardiovascular events, which is not captured by the ASCVD score. This is not a new finding but not an unimportant one, neither. The missed chance of the authors seems to be that they do not address a clinically urgent topic, i.e. to make a suggestions for the modification of the ASCVD risk score to incorporate cancer. (see Discussion in line 237 and line 279

Response 13: Thank you for stating that our data is interesting and that there is a point in our analysis. As per your suggestions above, we have now made efforts to be more cautious in the interpretation of our findings. We have also incorporated your suggestion to make suggestions for the modification of the ASCVD risk score to incorporate cancer.

“Hence, we propose that further research is needed to ascertain the value of adding cancer as a variable to the ASCVD score, as well as to identify and include relevant cancer-related variables to improve its applicability within patients with prior cancer.”

Round 2

Reviewer 2 Report

Thank you for abiding by my suggestions. In its current form, the study is acceptable. 

Minor issues: 

“We hypothesized that patients with prior cancer had a lower ASCVD score at incident AMI compared to those without cancer.”

Please provide the rationale for this statement, which you give in the reviewer comments. I did not find it in the text and therefore this statement appears unsupported.

Spelling error line 192

Author Response

Response to Reviewer 2 Comments

Point 1: Thank you for abiding by my suggestions. In its current form, the study is acceptable. 

Minor issues: 

“We hypothesized that patients with prior cancer had a lower ASCVD score at incident AMI compared to those without cancer.”

Please provide the rationale for this statement, which you give in the reviewer comments. I did not find it in the text and therefore this statement appears unsupported.

Response 1: Thank you for your constructive comments and positive feedback. We have made arrangements to the statements in the paragraph such that the rationale flows better as rightly pointed out.

“We have previously reported lower serum cholesterol concentrations, an important variable in existing risk scores including the ASCVD score, in patients with cancer compared with non-cancer controls at index AMI and stroke [13]. Hence, we hypothesized that patients with prior cancer would have a lower ASCVD score at incident AMI compared to those without cancer.”

Point 2: Spelling error line 192

Response 2: Thank you for highlighting this. We have made the amendments as below:

“Variation in predicted cardiovascular risk by cancer status”.
